# Characterization of a Hg^2+^-Selective Fluorescent Probe Based on Rhodamine B and Its Imaging in Living Cells

**DOI:** 10.3390/molecules26113385

**Published:** 2021-06-03

**Authors:** Wenting Zhang, Chunwei Yu, Mei Yang, Shaobai Wen, Jun Zhang

**Affiliations:** 1Laboratory of Environmental Monitoring, School of Tropical and Laboratory Medicine, Hainan Medical University, Haikou 571199, China; zhangwt2021@163.com (W.Z.); hy0211049@hainmc.edu.cn (C.Y.); wenshaobai@163.com (S.W.); 2School of Public Health, Hainan Medical University, Haikou 571101, China; myang_1995@163.com; 3Laboratory of Tropical Biomedicine and Biotechnology, Hainan Medical University, Haikou 571101, China

**Keywords:** Hg^2+^, fluorescent probe, rhodamine B, cell imaging

## Abstract

A small organic molecule **P** was synthesized and characterized as a fluorometric and colorimetric dual-modal probe for Hg^2+^. The sensing characteristics of the proposed probe for Hg^2+^ were studied in detail. A fluorescent enhancing property at 583 nm (>30 fold) accompanied with a visible colorimetric change, from colorless to pink, was observed with the addition of Hg^2+^ to **P** in an ethanol-water solution (8:2, *v*/*v*, 20 mM HEPES, pH 7.0), which would be helpful to fabricate Hg^2+^-selective probes with “naked-eye” and fluorescent detection. Meanwhile, cellular experimental results demonstrated its low cytotoxicity and good biocompatibility, and the application of **P** for imaging of Hg^2+^ in living cells was satisfactory.

## 1. Introduction

Mercury can exist in elemental, inorganic, and organic forms in the environment, among which Hg^2+^ is a carcinogenic and caustic material with high biological toxicity [1]. It can form methylmercury naturally by biomethylation in aquatic environments. As is known, this form of organic mercury is much more toxic than Hg^2+^, which can cause brain damage and other serious diseases [2,3]. Therefore, it is of great importance to develop efficient analytical methods to detect Hg^2+^ in the environment and biosystems. Over the past few years, different analytical methods, including electrochemical methods [4], inductively coupled plasma-mass spectrometry [5], and UV-Vis spectrometry [6], have been applied for the detection of Hg^2+^, but most of these methods are complicated, costly, and especially not suitable for in vitro/vivo applications. In recent years, the design of Hg^2+^-selective and sensitive fluorescent probes have attracted considerable interest due to the fact of their remarkable advantages such as low cost, operational simplicity, and non-destructiveness [7]. Though examples of “turn-on” Hg^2+^ probes have become available that display high selectivity and sensitivity for Hg^2+^ in micellar media and neutral aqueous samples [8,9], even imaging in zebrafish [10], most of the reported Hg^2+^-selective fluorescent probes are based on fluorescence quenching (“on–off”) mechanisms due to the spin–orbit coupling effect of Hg^2+^ [11,12,13], which is not favored over a fluorescence enhancement signal in light of selectivity and sensitivity concerns. Therefore, the synthesis of fluorescence enhancement (“off–on”)-type Hg^2+^-selective probes is still a challenge.

Rhodamine spirocyclic form derivatives are non-fluorescent and colorless, whereas strong fluorescence emission and a visible color change can be displayed upon combination with the targets. This recognition progress is caused by ring-opening of the corresponding rhodamine spirolactam [14]. This structural change has been widely used as a recognition mode to construct fluorescent and colorimetric probes for many analysts [15,16,17]. As to visualizing the subcellular distribution of metal ions in physiological processes, fluorescence imaging techniques have become a powerful tool [18]. Some Hg^2+^-selective fluorescent probes derived from rhodamine have been reported [19,20,21,22,23,24,25,26,27]. However, these reported rhodamine-based Hg^2+^-selective fluorescent probes still have shortcomings that need to be overcome, such as cross-sensitivities toward other metal ions and anions [27], pH dependency [28], and non-suitability for cell imaging [22,23], which could lower the sensitivity and limit the practical application of probes in environmentally and biologically relative targets. Compared with some successful fluorescence “turn-on” probes for imaging intracellular metal ions, such as Cu^2+^ [29], Al^3+^ [30], and Mg^2+^ [31], the development of highly selective, sensitive, and cell membrane-penetrable Hg^2+^ fluorescent probes with “off–on” signals is still a bottleneck. Therefore, ”off–on” fluorescent probes for Hg^2+^ based on rhodamine derivatives in environmental water samples and living cells are still a very active and significant challenge now and in the future. Benzoyl hydrazide derivatives have been extensively utilized to construct fluorescent probes in view of their remarkable optical properties. Moreover, benzoyl hydrazide derivatives are efficient selective receptors for the recognition of metal ions due to the multiple N and O binding sites [32,33,34], which effectively modulates their fluorescence. Accordingly, benzoyl hydrazide derivatives could play dual roles both as receptor units and reporters in probes.

In light of the abovementioned reasons, a 2-hydroxybenzoyl hydrazide-modified rhodamine derivative, **P**, was synthesized in this paper, and it was successfully characterized as a highly Hg^2+^ selective and sensitive fluorescent probe both in aqueous media and living cells. The synthesis route of proposed **P** is shown in Scheme 1.

## 2. Results and Discussion

### 2.1. pH Effect on the Fluorescent Response of ***P*** and a ***P***-Hg^2+^ System

The content of water in the testing system has a great effect on the response of the fluorescent probes, because the addition of water can lead to the precipitation of the probe that could cause a decrease in fluorescent intensity. In order to prevent this phenomenon, the volume ratio 8:2 of ethanol and water was adopted, which laid a foundation for the application of the probe **P** in the environmental samples. Real-time determination was necessary, and the time evolution of the responses of **P** (5 μM) in the presence of 10 equivalent of Hg^2+^ in the same buffer solution was also investigated (Appendix A); the recognition interaction was completed after the addition of Hg^2+^ within 15 min.

Subsequently, the effects of pH on the probe **P** and the **P**-Hg^2+^ system spectra were investigated (Figure 1). The pH had no obvious effect on the fluorescent spectra of **P**, which meant insensitive to acidity. For the **P**-Hg^2+^ system, the fluorescent intensity at 583 nm reached a maximum within the range 7.0–7.5, which is beneficial for use in biological systems. Thus, the followed fluorescent measurements were all conducted in the optimized conditions (ethanol-water, 8:2, *v*/*v*, 20 mM HEPES, pH 7.0).

### 2.2. Selectivity and Sensitivity Measurement of ***P***

A fluorescent probe with good selectivity was required for the detection of environmental or biological targets with complex backgrounds. The selectivity of this proposed probe, **P**, was conducted in an aqueous media (ethanol-water, 8:2, *v*/*v*, 20 mM HEPES, pH 7.0), and the tested metal ions were alkali, alkali-earth metals, divalent transition metal ions, including K^+^, Na^+^, Ag^+^, Ca^2+^, Mg^2+^, Zn^2+^, Pb^2+^, Cd^2+^, Ni^2+^, Co^2+^, Cu^2+^, Hg^2+^, Cr^3+^, and Fe^3+^, and the anions were Br^−^, I^−^, NO_3_^−^, H_2_PO_4_^−^, ClO_4_^−^, CO_3_^2^^−^, and SO_4_^2^^−^. The results showed that like most of the spirocycle rhodamine derivatives [19,20,21], the free **P** displayed a very weak fluorescence, which indicates that the spirolactam form was the predominant species. Introduction of the Hg^2+^ to probe **P** elicited an obviously fluorescent enhancement at 583 nm. By contrast, other metal ions and anions had almost no influence on the fluorescent spectra of **P** (Figure 2). Most likely, it was the addition of Hg^2+^ to **P** that caused the opening of the spirolactam in the structure of the rhodamine part [19,20,21,22,23,24,25,26,27], inducing an enhancement in fluorescence intensity. Furthermore, for the further study of the selectivity of **P**, a competition experiment was also conducted (Appendix A); all the tested metal ions and anions did not show any obvious interference to the response of **P** with Hg^2+^, except I^−^ had some influence on the response of **P**. This revealed that this proposed probe **P** could work in a complicated environment and has potential application in real samples. All these results also demonstrate that **P** could be employed as an Hg^2+^-selective probe.

### 2.3. Fluorescent and UV-Vis Titration Experiments of ***P*** to Hg^2+^

In order to further study the sensing properties and mechanism between **P** and Hg^2+^, fluorescence and absorption titration experiments in aqueous media (ethanol-water, 8:2, *v*/*v*, 20 mM HEPES, pH 7.0) were recorded (Figure 3). As the sequential introduction of Hg^2+^ to **P** (5 μM), the fluorescent intensity at 583 nm enhanced gradually, and the linear fluorescent intensity was proportional to the concentrations of Hg^2+^ in the range 1.0–20 μM with a detection limit of 0.33 μM (Appendix A). The UV-Vis spectra also gave similar results (Figure 3b); the maximum absorption peak at 560 nm appeared with increasing intensity upon successive addition of Hg^2+^, and a linear dependence of absorbance at 560 nm was observed as a function of Hg^2+^ concentration (inset of Figure 3b) in the range 2.0–20 μM. The association constant *K* was determined from the slope to be 3.18 × 10^4^ M^−1^ [35], corresponding to a stronger binding capability toward Hg^2+^ (Appendix A). The results showed that **P** was capable of detecting Hg^2+^, both qualitatively and quantitatively.

### 2.4. Coordination Mechanism of ***P*** with Hg^2+^

The stoichiometry of the **P**-Hg^2+^ complex was determined by a Job’s plot experiment in aqueous media (ethanol-water, 8:2, *v*/*v*, 20 mM HEPES, pH 7.0), and the total concentrations of **P** and Hg^2+^ was kept at 10 µM. When the mole ratio **P**/Hg^2+^ was at 1:1, the fluorescent intensity at 583 nm reached the maximum (Figure 4), which indicates that **P** coordinated with Hg^2+^ in a 1:1 stoichiometric relationship. Meanwhile, an experiment with Na_2_S as a competitive complexing agent could serve as experimental evidence to support this semi-reversible spiro ring-opening mechanism (Appendix A). To further explore the binding mode of **P** with Hg^2+^, the ^1^H-NMR spectra of **P** and **P**-Hg^2+^ in DMSO-*d_6_* were carried out (Appendix A). The proton peaks of –OH and O=C–NH in **P** alone existed in the form of hydrogen bonds and showed wide peaks and δ values that were somewhat larger than normal protons in the spectra of ^1^H-NMR, and the addition of Hg^2+^ to **P** in DMSO-*d_6_* solution led to a high-field shift of the signals –OH and O=C–NH at the degrees 0.0237 and 0.0069. It may be that the coordination of Hg^2+^ with **P** destroyed the formation of hydrogen bonds, the proton peaks of –OH and O=C–NH turned sharp, and the *δ* values became smaller than in **P**. According to the abovementioned results, **P** was most likely to chelate with Hg^2+^ via its oxygen on phenol hydroxylation, oxygen on the carbonyl group as well as nitrogen on the hydrazine. The proposed reaction mechanism of **P** with Hg^2+^ is shown in Scheme 2.

### 2.5. Preliminary Application of ***P*** in Cell Imaging

To further explore the biological applicability of **P** for Hg^2+^ in practical samples, intracellular Hg^2+^ imaging in HeLa cells by fluorescence microscopy was performed (Figure 5). Obvious fluorescence was not observed upon incubation with **P** (1.0 µM) for 30 min at 37 °C (Figure 5a), suggesting that autofluorescence from the cells could be avoided. Under the same testing conditions, stronger fluorescent change was detected after the addition of exogenous Hg^2+^ (1.0 µM) to the **P**-loaded HeLa cells (Figure 5b) which demonstrated that **P** could penetrate the cell membrane and coordinate with Hg^2+^ inside the cells. Moreover, brightfield imaging confirmed that the cells were viable after incubation with Hg^2+^ and/or **P** (Figure 5(a2,b2)). Meanwhile, **P** was also applied to the subcellular locations of Hg^2+^ in the HeLa cells using confocal fluorescence microscopy. The cells were co-treated with **P** (1.0 µM) and Hoechst 33342 (0.25 µg/mL) for 30 min with the same conditions as those used in Figure 5a,b. Our work revealed that the cellular localization and distribution of **P** was located primarily in the cytoplasm of those living HeLa cells as shown in Figure 5c. All the results indicated that the proposed **P** was an effective probe for imaging changes in Hg^2+^ intracellularly under biological conditions.

To evaluate the cytotoxicity of the fluorescent probe **P** in living cells, an MTT assay on PC12 cells with **P** concentrations from 0 to 10 µM was taken (Appendix A). After treatment with **P** for 48 h, the cellular viability was estimated at approximately 92%, which exhibited the low toxicity of **P** to cultured cells.

## 3. Materials and Methods

### 3.1. Main Reagents and Instruments

All reagents were of commercially analytical grade and used directly.

Fluorescent spectra were recorded using a Hitachi F-4600 spectrofluorometer (Tokyo, Japan), and UV-Vis spectra were determined on a Hitachi U-2910 spectrophotometer. ^1^H- and ^13^C-NMR spectra were carried out with a Brucker AV 400 nuclear magnetic resonance instrument (Faellanden, Switzerland), and the chemical shift is given in ppm from tetramethylsilane (TMS). Mass spectra were obtained using a thermo TSQ Quantum Access Agilent 1100 mass spectrometer (Santa Clara, CA, USA). Fluorescence imaging was performed with Olympus FluoView Fv3000 laser scanning microscope (Tokyo, Japan).

### 3.2. Synthesis of Probe ***P***

Compounds RBH and RBHO were obtained according to the reported method [36].

An amount of 0.1521 g of 2-hydroxybenzoyl hydrazide (1.0 mmol) and 0.4964 g RBHO (1.0 mmol) were dissolved in ethanol (40 mL) and added to a round-bottom flask (100 mL). The mixture was reacted under reflux for 6 h, and then cooled to room temperature, and the yellow precipitate so obtained was filtered off and washed with cold ethanol. Yields: 85.6%. MS *m*/*z*: 631.5 [M + H]^+^, 629.5 [M − H^+^]^−^. ^1^H-NMR (DMSO-*d_6_*, δ ppm): 11.82 (s, 1H), 11.38 (s, 1H), 7.98 (d, 1H, *J* = 8.16), 7.92 (d, 1H, *J* = 7.40), 7.88 (d, 1H, *J* = 8.16), 7.70 (d, 1H, *J* = 6.80), 7.61 (t, 1H, *J* = 8.00), 7.57 (t, 1H, *J* = 6.00), 7.40 (t, 1H, *J* = 8.36), 7.06 (d, 1H, *J* = 7.52), 6.93 (t, 1H, *J* = 7.20), 6.89 (s, 1H), 6.45 (s, 1H), 6.44 (d, 3H, *J* = 7.28), 6.36 (d, 2H, *J* = 8.00), 3.32 (m, 8H, *J* = 8.70), 1.07 (t, 12H, *J* = 6.98). ^13^C-NMR (DMSO-*d_6_*, δ ppm): 164.88, 164.77, 158.77, 152.65, 152.53, 149.13, 147.56, 143.55, 135.02, 134.29, 129.40, 127.80, 127.42, 124.17, 123.79, 119.48, 117.57, 117.05, 108.67, 104.71, 97.93, 65.43, 56.50, 44.13, 19.04, 12.89. (Appendix A Appendix A, Appendix A)

### 3.3. General Spectroscopic Methods

The stock solution (1.0 mM) of **P** was obtained by dissolving **P** with DMSO. All the salts solutions (1.0 mM) were obtained by dissolving in deionized water. Before fluorescent and UV-Vis spectroscopic measurements, all the testing solutions were obtained by diluting the stock solutions to the desired concentration. The excitation and emission monochromator slit widths of the fluorescence spectrophotometer were 10 nm and 10 nm, respectively, and the excitation wavelength was fixed as 530 nm.

### 3.4. Cell Incubation and Imaging

HeLa cells were placed on coverslips and washed with PBS (phosphate-buffered saline). It was then incubated with 1.0 μM **P** (dissolved with DMSO) for 30 min at 37 °C, followed by washing three times with PBS. The cells were further cultured with 1.0 μM of HgCl_2_ for 30 min at 37 °C and washed with PBS three times again. The fluorescence cell imaging of intracellular Hg^2+^ in HeLa cells was conducted by a confocal fluorescence microscopy on an Olympus FluoView Fv1000 laser scanning microscope.

## 4. Conclusions

In summary, a highly selective Hg^2+^ probe, **P**, derived from rhodamine B was designed and investigated by fluorescence and UV-Vis techniques. On the basis of the change between spirolactam and open-cycle forms in the rhodamine unit, the sensing mechanism of this proposed probe was studied in detail. Furthermore, confocal fluorescence microscopy experiments demonstrated that probe **P** can be applied to image Hg^2+^ in living cells.

## Data Availability

Data supporting the reported results are available online.

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
