# Peer review of "Characterization of a Hg2+-Selective Fluorescent Probe Based on Rhodamine B and Its Imaging in Living Cells"

_molecules, 2021, doi:10.3390/molecules26113385_

Round 1
Reviewer 1 Report
1) Page 4, line 136: in Figure 1, X axisis badly marked. Replace „wavelength, nm“ with pH. The measurement was in ethanol-water 8: 2 or ???
2) Figure S5: Fluorescence of P at 530 nm excitation must be 0. P does not absorb light at 530 nm !!!The presence of water will not affect it.The fluorescence signal in pure water (Figure S5) is equal to 0.In my opinion, the monotonic decrease of the fluorescence signal (Figure S5) is caused by the precipitation (insolubility) of P at a higher amount of water in solution.UV-Vis measurements in the range of 300 to 450nm would confirm this.It is necessary to check this and discuss the effect of water on the fluorescence of P, resp. P + Hg2+ and Figure S5 must be removed from the work.The ethanol-water mixture is not a suitable medium for this application.This is also confirmed by the results of the P in cell imaging application.I recommend using DMSO-water for the measurements. It will be possible to measure at higher volumes of water in solution.This will also increase the quality of measurements on a confocal microscope and the experimental conditions will be more similar to real conditions.
3) In Figures S6 a) and b), it is necessary adjust the Y axes to have the same range; mark a) insert into the figure.
4) It is necessary to set a detection limit for the determination of Hg2 + for the fluorescence and for the absorption measurements.Without these data, it is not possible to compare the sensitivity of P with the previously published results of other organic compounds that have been described in the literature as probes or Hg2+ sensors.The P + Hg2+ absorption spectrum strongly overlaps with the P+ Hg2+ fluorescence spectrum Figure 3.At absorbances higher than 0.1, the titration fluorescence curve will be affected by the P + Hg2+ concentration, which will affect the sensitivity of the probe.
Author Response
The authors are very grateful to reviewer for the valuable comments. The manuscript has been carefully revised in light of the comments. The full-text, Schemes and Figures are all carefully checked and corrected for improvement. The revised sections have been marked in red in the revised manuscript.

Reviewer 2 Report
The manuscript describes the development of a fluorogenic and colorimetric Hg2+ indicator. The compound is based on the Rhodamine spirolactam framework, which has been extensively studied to prepare turn-on type fluorescent sensors for divalent cations, including Hg2+. This study does not meet the criteria for publication at this stage because the following points have not been convincingly clarified.
(1) The significance of the study is unclear compared to previous studies led by other groups. The authors note that few studies have been reported on fluorescent Hg2+ sensors that are fluorogenic (turn-on type), unaffected by pH, and can be used in living cells. However, several studies on fluorogenic Hg2+ sensors with high selectivity have already been reported, which are not cited in this manuscript. Some can be used in neutral aqueous solutions (J. Am. Chem. Soc. 2004, 126, 8, 2272–2273, Org. Lett. 2007, 9, 1, 121–124), unaffected by pH ranges at 5-10 (Org. Lett. 2007, 9, 5, 907–910), and others were used in living cells and even in zebrafish (J. Environ. Monit., 2009, 11, 330-335, Org. Lett. 2010, 12, 3, 476–479, Org. Lett. 2012, 14, 3, 820–823, ). Some of them include Rhodamine 6G spirolactam skeletons, which are similar to sensor P in this study, especially the chemical structure of Rho-Hg1 reported by Peng’s group (Org. Lett. 2010, 12, 3, 476–479) is very close to P. At least, this paper must be cited.
(2) The coordination mechanism is only a speculation. No convincing experimental data have been indicated. 1H NMR spectra of P (Fig S3) and P+Hg2+ (Fig S7) are almost the same. If the P/Hg complex has the open form structure proposed in Scheme 2, the chemical shifts in the aromatic region must differ from those of P. It is strange.
(3) There is no time course for the increase in fluorescence intensity when Hg2+ is added. Also, no data has been presented to show whether this process is reversible or not.
(4) P seems to have a solubility problem in water because all the experiments other than live-cell experiments were conducted in 80% ethanol (Figures 1-3). In addition, the fluorescent intensity of P/Hg2+ sharply decreases with increasing concentration of water (Fig S5). P/Hg2+ is not fluorescent in pure water. These should interfere with the live-cell applications.
(5) The authors do not provide sufficient data to show the selectivity of mercury ions over other metal ions. For all the metal ions tested, the fluorescence intensity of P in the presence of the same concentration of the metal ion should be shown.
(6) In the inset in Figure 3, the fluorescence intensity and [Hg2+] do not show a linear relationship. This can be an essential drawback for sensors intended to quantify an analyte.
Author Response

(The authors gave the same response as above.)

Reviewer 3 Report
The manuscript by Zhang et al., characterizes a Rhodamine B based Hg2+ selective probe. In their studies they utilized absorption spectroscopy as well as both in vitro steady-state fluorescence and fluorescence live cell imaging microscopy. The motivation for this work is clearly stated since mercury, in several forms, is carcinogenic and highly toxic. One of the challenges in the development this probe is the requirement for high specificity for Hg2+. To this end, the authors synthesized a 2-hydroxybenzoyl hydrazide modified rhodamine B derivative, which proved to be both highly selective and sensitive to Hg2+ both in aqueous media and in living cells. The synthetic route utilized as well as the spectroscopic characterization were clearly delineated. The ion selectivity was tested against an impressive array of ions including the cations K+, Na+, Ag+, Ca2+, Mg2+, Zn2+, Pb2+, Cd2+, Ni2+, Co2+, Cu2+, Hg2+, Cr3+ and Fe3+ and the anions Br-, I-, NO3-, H2PO4-, ClO4-, CO32-, and SO42-. The fluorescence enhancement upon interaction with Hg2+ is impressive and their probe should prove to be highly useful. The imaging data are rather preliminary but serve to demonstrate that the probe can be taken up by live cells and that it can then report on intracellular Hg2+ levels. As regards the science presented and the results demonstrated I am very favorably inclined towards publication of this manuscript.
There are, however, several issues that should be addressed before publication.
Minor issues:
In line 58, the authors state:
“Rhodamine derivatives are non-fluorescent and colorless …”
I have personally worked with rhodamine compounds for more than 30 years and they are certainly not “non-fluorescent and colorless”. It is true that if the rhodamine compound exists in a spirolactam form it will be colorless until the ring opens. Perhaps the authors mean to say that their particular compound remains in the spirolactam form until it binds to Hg2+. Certainly their blanket statement about rhodamines in general is not correct and the authors should reconsider this claim.
Figure 1 appears to have a problem with the X-axis.
Major issue:
Although I am favorably impressed by the quality of the work overall I am concerned about the many errors in the English usage. The manuscript would be much better received by readers if the authors can have a native English speaker do a careful correction. Although bothersome, this step will certainly help the authors in the long run.
Author Response
The authors are very grateful to the reviewers for their valuable comments. The manuscript has been carefully revised in light of the comments. And the full-text, Schemes and Figures are all carefully checked and corrected for improvement. The revised sections have been marked in red in the revised manuscript.

Round 2
Reviewer 2 Report
Most of the questions and comments were answered. The revised manuscript has been improved. However, the concern about the selectivity of probe P remains. In my original comment (5), I wrote “The authors do not provide sufficient data to show the selectivity of mercury ions over other metal ions. For all the metal ions tested, the fluorescence intensity of P in the presence of the same concentration of the metal ion should be shown.” The authors’ answer is not satisfactory. Figure S6 shows whether the fluorescence intensity of the P-Hg complex is affected in the presence of other cations. This does not mean that the fluorescence intensity of probe P does not increase in response to other metal cations. Please add the data showing that probe P can selectively detect Hg2+ over other cations, as shown in Figure 1b of Reference 20 (Org. Lett. 2010, 12, 476-479), for example. Figure 2 shows the fluorescence spectra of P in the presence of the other tested cations (as noted in the figure), so the authors should already have that data.
Author Response
The authors are very grateful to the reviewer for the valuable comments. The manuscript has been carefully revised in light of the comments.
